# A generic pixel-to-point comparison for simulated large-scale ecosystem properties and ground-based observations: an example from the Amazon region

Anja Rammig[1,*], Jens Heinke[2,*], Florian Hofhansl[3], Hans Verbeeck[4], Timothy R. Baker[5], Bradley Christoffersen[6], Philippe Ciais[7], Hannes De Deurwaerder[4], Katrin Fleischer[1], David Galbraith[5], Matthieu Guimberteau[7], Andreas Huth[8], Michelle Johnson[5], Bart Krujit[9], Fanny Langerwisch[2], Patrick Meir[10,11], Phillip Papastefanou[1], Gilvan Sampaio[12], Kirsten Thonicke[2], Celso von Randow[12], Christian Zang[1], Edna Rödig[8]

[1]Technical University of Munich, TUM School of Life Sciences Weihenstephan, Hans-Carl-von-Carlowitz-Platz 2, 85356 Freising, Germany, Anja.Rammig@tum.de
[2]Potsdam Institute for Climate Impact Research, Potsdam, Germany
[3]IIASA International Institute for Applied Systems Analysis; Schlossplatz 1, A-2361 Laxenburg, Austria
[4]CAVElab Computational & Applied Vegetation Ecology, Department of Applied Ecology and Environmental Biology, Faculty of Bioscience Engineering, Gent, Belgium
[5]School of Geography, University of Leeds, Leeds UK
[6]Department of Biology, The University of Texas Rio Grande Valley, Edinburg, USA
[7]Laboratoire des Sciences du Climat et de l'Environnement, LSCE/IPSL, CEA-CNRS-UVSQ, Université Paris-Saclay, Gif-sur-Yvette, France
[8]Helmholtz Centre for Environmental Research (UFZ), Leipzig, Germany
[9]ALTERRA, Wageningen-UR, Wageningen, The Netherlands
[10]School of Geosciences, University of Edinburgh, Edinburgh, UK
[11]Research School of Biology, Australian National University, Canberra, Australia
[12]INPE, Sao Jose dos Campos, SP, Brazil
* both authors contributed equally to the manuscript.

*Correspondence to*: Anja Rammig (Anja.Rammig@tum.de)

**Abstract.** Comparing model output and observed data is an important step for assessing model performance and quality of simulation results. However, such comparisons are often hampered by differences in spatial scales between local point observations and large-scale simulations of grid-cells or pixels. In this study, we propose a generic approach for a pixel-to-point comparison and provide statistical measures accounting for the uncertainty resulting from landscape variability and measurement errors in ecosystem variables. The basic concept of our approach is to determine the statistical properties of small-scale (within-pixel) variability and observational errors, and to use this information to correct for their effect when large-scale area averages (pixel) are compared to small-scale point estimates. We demonstrate our approach by comparing simulated values of aboveground biomass, woody productivity (woody net primary productivity, NPP) and residence time of woody biomass from four dynamic global vegetation models (DGVMs) with measured inventory data from permanent plots in the Amazon rainforest, a region with the typical problem of low data availability, potential scale mismatch and thus, high model uncertainty. We find that the DGVMs under- and overestimate aboveground biomass by 25% and up to 60%, respectively.

Our comparison metrics provide a quantitative measure for model-data agreement and show moderate to good agreement with the region-wide spatial biomass pattern detected by plot observations. However, all four DGVMs overestimate woody productivity and underestimate residence time of woody biomass even when accounting for the large uncertainty range of the observational data. This is because DGVMs do not represent the relation between productivity and residence time of woody

biomass correctly. Thus, the DGVMs may simulate the correct large-scale patterns of biomass but for the wrong reasons. We conclude that more information about the underlying processes driving biomass distribution are necessary to improve DGVMs. Our approach provides robust statistical measures for any pixel-to-point comparison, which is applicable for evaluation of models and remote sensing products.

## 1 Introduction

The rate of environmental change in tropical South America and in particular in the Amazon region has been unprecedented in the last decades (e.g. Lewis *et al.* 2011; Davidson *et al.* 2012). Estimates of the amount of carbon stored in tropical rainforest biomass differ strongly (Avitabile et al., 2016;Baccini et al., 2012;Saatchi et al., 2011;Mitchard et al., 2014). In addition, estimated carbon release to the atmosphere from land-use change is uncertain (e.g. Houghton et al., 2012;Baccini et al., 2017;Song et al., 2015;Harris et al., 2012). Nonetheless, a successful implementation of protection incentives, e.g. for reducing

emissions from deforestation and degradation (REDD+), requires both accurate estimates of existing regional carbon stocks as well as improved projections of future scenarios (e.g. Langner et al., 2014).

Dynamic global vegetation models (DGVMs) are important tools to estimate impacts of climate and land-use change on the carbon cycle (e.g. Cramer *et al.* 2004; Sitch *et al.* 2008). To correctly capture carbon dynamics in tropical forests, DGVMs need to improve the simulation approach of drought-related mortality and other types of tree mortality that control stand density

(Pillet *et al.* 2018), they need to include more detailed gap dynamics influencing stand dynamics (Espírito-Santo et al., 2014;Rödig et al., 2017), and they need to incorporate how nutrient availability limits woody productivity (Quesada *et al.* 2012; Johnson *et al.* 2016). At the same time, model evaluation based on available data is necessary for which the primary source are ground-based observations of AGB obtained from forest census data (Lopez-Gonzalez et al. 2011; Brienen et al. 2014; Lopez-Gonzalez et al. 2014; Mitchard et al. 2014; Brienen et al. 2015; Johnson et al. 2016). When conducting data-

model comparisons at the plot scale, the spatial resolution of both get into focus. The size of forest plots is typically in the order of 1 ha or less, whereas average DGVM grid-cell resolution is determined by the available gridded climate data set, which is usually about several thousand square kilometers (>100,000 ha). Plot observations are affected by observational errors, uneven spatial distribution (Saatchi et al. 2015) and spatial variability due to natural gap dynamics (Chambers *et al.* 2013), and thus are likely to exhibit substantial deviation from average large-scale properties. The problem of comparing point

data with model results obtained at grid-cell (pixel) size occurs in many applications of remote sensing and ecological modelling.

So far, we lack a reliable and objective method to compare simulation results from DGVMs at grid-cell scale (pixel) and plot (point) observations. Several studies, that evaluated patterns of interpolated maps from plot data and model simulations, concluded that the observed and simulated spatial patterns do not match (e.g. Johnson *et al.* 2016). Here, we complement these findings by providing quantitative statistical measures for such comparisons and present an approach for performing pixel-to-

point comparisons while accounting for different statistical properties of point and pixel values and its uncertainties. The basic concept of the approach is to determine statistical properties from small-scale variability and observational errors in ecosystem variables, in order to account for these effects when comparing large-scale area averages (pixels) and small-scale plot estimates (points).

We apply our approach by comparing point estimates of ecosystem properties obtained from forest inventories (Mitchard *et*

*al.* 2014; Brienen *et al.* 2015) to corresponding simulated pixel values from four state-of-the-art DGVMs. Similar to Johnson et al. (2016), we focus on three ecosystem properties that are well defined and represented in both, DGVMs and forest inventories: (i) aboveground biomass (AGB, in Mg C ha$^{-1}$); (ii) aboveground woody productivity (WP, in Mg C ha$^{-1}$ yr$^{-1}$); and (iii) residence time of woody biomass ($\tau$ in years).

We evaluate the accuracy of the spatial pattern of these three ecosystem properties and provide statistical measures for the

quality of model simulations in comparison to observations, thereby accounting for small-scale landscape variability and associated measurement errors. We demonstrate the strength of our approach by highlighting its applicability for model evaluation and model benchmarking.

In particular, we address three research questions:

1) How well do models represent variations in aboveground biomass across the Amazon region? We expand the pure

20        (visual) qualitative comparison by deriving three statistical metrics for a quantitative comparison.

2) How can we evaluate differences between observed and simulated spatial biomass patterns (based on the presented metrics), in particular when considering different allometric equations? We discuss the effects of inadequate data and associated model uncertainty.

3) What can we learn from the spatial heterogeneity and underlying drivers of spatial biomass patterns? We analyze

25        simulated and observed patterns of WP and $\tau$.

## 2 Methods

Landscape variability depends on the extent and heterogeneity of the study area (Turner et al., 2001). Point measurements within a pixel of larger spatial scale, for example, may reveal small-scale spatial variabilities within the pixel. We derive a "within-pixel variability" that so far has not been accounted for in earlier approaches. We present three steps to calculate three

metrics that provide a measure on the best achievable correlation between point and pixel values (see Fig. 1).

## 2.1 A generic method for point-to-pixel comparisons

### 2.1.1 Calculate the "global variability" across the region of interest

Assume, we have a dataset $X$ with a number $N$ of point observations $x_i$ at location $i$ (e.g. plot observations from inventory data). In the first step, we calculate the mean $\bar{x}$ and variance $\sigma_x^2$ across all plots in a region (e.g. across the Amazon region).
The variance $\sigma_x^2$ denotes the global variability (i.e. the variability across the whole Amazon region) at point scale (Fig. 1a).

### 2.1.2 Calculate within-pixel variability

In the second step, we identify within-pixel variability from point measurements. With coarser pixel resolution, the spatial variability (here: global variability) is reduced. In order to compare pixel values against point values, global variability at point scale needs to be reduced by the within-pixel variability (variability component $\varepsilon_i$; Fig. 1b).
The variability component $\varepsilon_i$ is assumed to be normally distributed with zero mean and variance $\sigma_\varepsilon^2$:
$$\varepsilon_i \sim \mathcal{N}(0, \sigma_\varepsilon^2) \tag{1}$$
Based on the variability component $\varepsilon_i$, we estimate the within-pixel variance $\sigma_\varepsilon^2$ from the point observations by analysing their covariance, which is equivalent to the nugget effect (i.e. the sum of variance caused by small-scale variability and observation error) in a semivariogram (see SI Methods). Due to the limited amount of inventory data, we assume here, that $\sigma_\varepsilon^2$
is stationary across the region of interest (for details on that assumption see discussion and SI Results).

The global variance at point scale $\sigma_x^2$ now differs from the corrected global variance at pixel scale, $\sigma_{x,corr}^2$, as variances add quadratically, assuming that the (small scale) variability component $\varepsilon_i$ has errors uncorrelated to the global distribution of $x$ :
$$\sigma_{x,corr}^2 = \sigma_x^2 - \sigma_\varepsilon^2 \tag{2}$$

### 2.1.3 Metrics for the comparison of two datasets with different spatial resolutions

In a third step, we compare the point data $x_i$ with simulated data $y_i$ at pixel scale. Similar to the above procedure, we calculate the mean $\bar{y}$ and variance $\sigma_y^2$ for the simulated pixels that contain point observations (hereby we assign each point observation the pixel value in which the point is located). We then compare the simulation results by applying three metrics:

1) Mean bias (MB): the ratio of means $\bar{y}/\bar{x}$ across the whole region as a measure of the mean bias in the patterns which is not affected by small scale variability;

2) Pattern amplitude (PA): the ratio of standard deviations $\sigma_y/\sigma_{x,corr}$ using the corrected global variability (i.e. removed within-pixel variability) and serves as a measure of differences in pattern amplitude or in the variability of the simulated and observed data;

3) Similarity of pattern (SP): We use $r_{corr}$ as a measure of the similarity of the 'shape' of spatial patterns, i.e. the spatial correlation of simulated and observed data (see SI). Accordingly, we can calculate the maximum achievable

correlation coefficient $r_{max}$, which is derived from correlating the observational data set at point scale to the same observational data set at pixel scale (see Fig. 1a, b and SI).

The limited number of point observations and their non-random spatial distribution in the Amazon region affects the accuracy of the comparison. We therefore estimate confidence intervals for the comparison metrics MB, PA and SP, respectively, by applying a bootstrapping technique (10,000 repetitions). Because the estimation of $\sigma_\varepsilon^2$ is based on the analysis of the spatial correlation structure of the data, a block-bootstrapping is performed (Politis and Romano, 1994). For each permutation, the domain of observations is randomly divided into 100 tiles (random orientation and offset, ca. pixel size) from which a random recombination is drawn with replacement. This technique assures that the spatial correlation structure of the data remains intact.

## 2.2 Application of the pixel-to-point comparison to simulated and observed data from the Amazon region

### 2.2.1 Observed data at point scale: Description of site-level data

The observed data at point scale are forest census-based plot measurements across the Amazon region, in which all plots that were subject to anthropogenic disturbances, including selective logging were excluded (Brienen et al. 2015). The average plot size is ~1.2 ha (Brienen et al. 2015) so that the plots incorporate most size classes of natural gaps, particularly as the plot data were averaged across sites occurring within the same pixel. Across the plot network, the biases introduced into estimate of carbon balance by one hectare plots not sampling the very largest and rarest natural gaps are in fact very small (Espirito-Santo et al. 2014). We use datasets of AGB (Lopez-Gonzalez et al., 2011;Lopez-Gonzalez et al., 2014;Mitchard et al., 2014), WP and woody loss (WL; Brienen et al., 2014). WP and WL are derived "[…] from the sum of biomass growth of surviving trees and trees that recruited (that is, reached a diameter ≥ 100 mm), and mortality [=woody loss] from the biomass of trees that died between censuses" (Brienen et al. 2015). We convert AGB, WP and WL from dry biomass to carbon mass (see SI Methods). For the calculation of AGB, we use different allometric equations that account e.g. for regional differences in wood density or tree height (Table S1). We exemplify our comparison metrics based on AGB calculated from the three parameter moist tropical forest allometry from Chave et al. (2005), where tree height is estimated from DBH individually for each stem based on the region-specific Weibull models from Feldpausch et al. (2012). Wood density is estimated for each stem using the mean value for the species in the Global Wood Density Database (Chave et al., 2009;Zanne et al., 2009), or the mean for the genus using congeneric taxa from Mexico, Central America and tropical South America if no data were available for that species (Mitchard et al. 2014). We here evaluate the principal AGB dataset ($K_{DH\rho}$) from Mitchard et al. (2014) in more detail (the other allometric equations are presented in the supplement).

### 2.2.2 Simulated data at pixel scale: Description of DGVM simulations

We use outputs from four state-of-the-art DGVMs, namely the Lund-Potsdam-Jena DGVM for managed Land (LPJmL, Bondeau et al., 2007;Gerten et al., 2004;Sitch et al., 2003), the Joint U.K. Land Environment Simulator (JULES), v. 2.1. (Best

et al., 2011;Clark et al., 2011), the INtegrated model of LAND surface processes (INLAND) model (a development of the IBIS model, Kucharik et al., 2000) and the Organising Carbon and Hydrology In Dynamic EcosystEms (ORCHIDEE) model (Krinner et al., 2005). A short description of each of the applied models is provided in the Supplementary Information (SI Methods). The models were applied to the Amazon region covering the area of 88°W to 34°W and 13°N to 25°S at a spatial

resolution of 1°x 1° lat/lon. The resolution of the DGVMs is defined by the resolution of the climate input data for which we here used bias-corrected NCEP meteorological data (Sheffield et al., 2006). Model runs were performed based on the standardized Moore Foundation Andes-Amazon Initiative (AAI) modelling protocol (Zhang et al., 2015). The same set of models and output variables was analysed in Johnson et al. (2016).

### 2.2.3 Comparing inventory and simulation results

In our application, data set $X$ corresponds to the inventory measurements at point scale (Fig. 1a). For this dataset, we have to derive the within-pixel variability (Fig. 1b). Data set $Y$ corresponds to the simulated pixel values (Fig. 1c). Hence, the pixel scale is defined by the resolution of the model simulation (1°x1°, approximately 12,200 km², Fig. 1c). We calculate the three metrics (sect. 2.1.3) from the observed and simulated ecosystem variables AGB, WP and τ.

## 3 Results

**3.1 Comparison of aboveground biomass (AGB)**

The visual comparison indicates that the spatial pattern of AGB from the plots (Fig. 2a and Fig. S1) differs from the spatial patterns of AGB simulated by either DGVM (Fig. 2c-f). In addition, the DGVM patterns are vastly different among each other. Mean $\bar{x}$ and global variability $\sigma_x$ of AGB for all plot observations across the Amazon region (Fig. 3a) range from 134-153 and 36-50 MgC ha$^{-1}$, respectively (Tab. S2), depending on the allometric equation applied. Within-pixel variability $\sigma_\varepsilon$, as calculated

from Eq. S1, ranges between 28 and 36 MgC ha$^{-1}$. The corrected variability of observed AGB at pixel-scale ($\sigma_{x,corr}$) is thus substantially lower than the global variability and ranges between 22-39 MgC ha$^{-1}$ (Fig. 4a, Tab. S2). Based on these estimates we calculate the maximum achievable coefficients $r_{max}$ for a comparison between pixel averages and point estimates of 0.61 to 0.78 for different allometric equations (Fig. 5).

The models simulate a continuous cover of biomass across the Amazon region at a spatial resolution of 1°x1° pixel size. For

our comparison, we only use the simulated pixel values of AGB at each plot location. Thus, the estimated statistical properties are not representative for the entire Amazon region but only for a relatively small subset of pixels (i.e. 98 pixels as in Fig. 2a/b). For simulated AGB from the four DGVMs, we estimate a mean $\bar{y}$ of 114 MgC ha$^{-1}$ for INLAND, 151 MgC ha$^{-1}$ for JULES, 217 MgC ha$^{-1}$ for ORCHIDEE and 170 MgC ha$^{-1}$ for LPJmL (Fig. 3a, Tab. S3). Depending on the allometric equation applied to calculate observed biomass (Tab. S1), INLAND underestimates mean AGB by 15-25%. LPJmL and ORCHIDEE

overestimate AGB by 11-26 and 42-62%, respectively. JULES deviates only by 1% from AGB derived from the 2-parameter

allometric equations (kdr2p, kd2p), but overestimates AGB derived from all 3-parameter allometric equations by 12% (Tab. S3).

Mean global variability of simulated AGB, $\sigma_y$, ranges between 13 MgC ha$^{-1}$ for JULES and 62 MgC ha$^{-1}$ for ORCHIDEE (Fig. 4a and Tab. S3). Without correcting for small-scale variability $\sigma_\varepsilon$ in the point-to-pixel comparison, we would conclude that

the pattern amplitude simulated by ORCHIDEE and LPJmL agree quite well with observed patterns (Fig. 4b). However, when accounting for the lower corrected variability ($\sigma_{x,corr}$), because the error of observation-based estimates at pixel-level is smaller, it becomes easier to falsify models with uncertain data. We find that LPJmL and ORCHIDEE both overestimate the observed spatial amplitude by 43% and 62%, respectively (Fig. 4c and see Tab. S3 for other allometric equations). For INLAND and JULES, on the other hand, we find a corresponding underestimation of pattern amplitude by 14 and 65%,

respectively. We also note that confidence intervals for $\sigma_y/\sigma_{x,corr}$ are large in particular for ORCHIDEE and LPJmL (Fig. 4c).

Correlation coefficients indicating the similarity of simulated and observed patterns of AGB range from 0.25 - 0.53 (corrected) across all models (Tab. S3). The highest similarity of pattern (i.e. best correlation values $r_{corr}$) is found for ORCHIDEE, lowest similarity of pattern for LPJmL. Across the three models INLAND, JULES and LPJmL, generally, higher similarity of pattern

is found for the allometric equations that include regional height models and mean or species specific wood density (K$_{DH\rho}$, K$_{DH}$; Fig. 5).

### 3.2 Comparison of woody productivity (WP)

Mean $\bar{x}$ and variability at pixel-scale $\sigma_{x,corr}$ of observed WP are 2.57 and 0.38 Mg C ha$^{-1}$ yr$^{-1}$, respectively. There seems to be a weak spatial pattern in the plot estimates at pixel level (Fig. 6a), which is not reflected by the models (Fig. 6c-f). The DGVMs

display a distinct pattern of WP across the region that strongly differs among the four models (Fig. 6c-f).

Mean WP simulated by the DGVMs ($\bar{y}$) is between 4-5 Mg C ha$^{-1}$ yr$^{-1}$ for LPJmL and JULES, and 8-9 Mg C ha$^{-1}$ yr$^{-1}$ for INLAND and ORCHIDEE, respectively (Tab. S4). All DGVMs strongly overestimate mean WP (Tab. 1A). In addition, most models overestimate the pattern amplitude, and the simulated variability ranges between 0.72 and 1.6 Mg C ha$^{-1}$ yr$^{-1}$ (Tab. S4). Pattern similarity of observed and simulated data is low ranging from 0.03 to 0.50 (Tab.1A), even with a relatively low

maximum achievable correlation of 0.65 (Tab. 1A).

### 3.3 Comparison of residence time of woody biomass ($\tau$)

Mean $\bar{x}$ and variability at pixel-scale $\sigma_{x,corr}$ of observed $\tau$ are 74 and 28 years, respectively. Again the visual comparison shows that the simulations do not match the observations (Fig. 7a vs. Fig. 7c-f). The simulated mean $\bar{y}$ of $\tau$ ranges between 15 (INLAND) to 35 (LPJmL) years with a variability of 3 (INLAND) to 8 (LPJmL) years. This is displayed in our comparison

metrics: Mean bias results in very low values (i.e. strong underestimation of 53 to 80%; Tab. 1B) and pattern amplitude is strongly underestimated by 65 to 87% (Tab. 1B). The similarity of pattern is very low for all models (Tab. 1B).

## 4 Discussion

We here present a novel approach for a pixel-to-point comparison. We account for the reduced observed variability when going from point to pixel scale by evaluating three indicators, i.e. the mean bias, the pattern amplitude and the similarity of spatial pattern (section 1.3). We use an example from the Amazon region by comparing model output from four DGVMs and forest inventory data. In the following, we discuss our findings of substantial discrepancies between simulated and observed patterns of AGB, WP and τ across the Amazon region.

### 4.1 How well do model simulations represent observed biomass patterns across the Amazon?

Interpolated biomass maps from plot observations (e.g. Johnson et al., 2016;Malhi et al., 2006) should be treated with caution since plot observations may not be representative at the landscape-scale (Chave et al., 2004). As a result, a direct and meaningful comparison of observed and simulated maps is currently not feasible but reliable biomass estimates are necessary for implementation of protection incentives and future projections of vegetation biomass. Our results demonstrate that most models are in good agreement and deviate from mean observational biomass by less than 20% (i.e. low mean bias c.f. Fig. 3) and their variability at landscape-scale deviates by about 40% (i.e. pattern amplitude, c.f. Fig. 4). Such relatively good agreement was also found in simulation runs from Delbart et al. (2010) and Johnson et al. (2016). Our results even yield relatively high similarity in observed and simulated spatial patterns of AGB at pixel scale (except LPJmL; c.f. Fig. 5), given the fact that the maximum achievable correlation in the data itself is only 0.6-0.8 (Fig. 5). This indicates that there is considerable uncertainty in the data, which needs to be considered in point-to-pixel comparisons and which we elaborate in the following paragraph.

### 4.2 How to evaluate differences between observed and simulated patterns of biomass (based on the presented metrics), in particular when considering different allometric equations?

As discussed by several authors (e.g. Baker et al., 2004;Chave et al., 2006;Chave et al., 2014;Réjou-Méchain et al., 2017), the methodology used to convert plot measurements to actual biomass may lead to differential biomass estimates depending on the respective assumptions of the allometric equations employed (i.e. using species-level or community mean wood density, and region-specific or basin-wide height models, see SI Tab. S1). As a result, we here find a more or less pronounced pattern of biomass variability across the Amazon region based on respective assumption used (c.f. Fig. 2, Fig. S1). While mean global variability of biomass is highest for the allometric equations including species-level wood density (c.f. Figure S1, Table S2), highest within-pixel variability is found for biomass values estimated from two-parameter allometric equations (Tab. S2) excluding tree height (c.f. Table S1). This result is also reflected by the lower maximum achievable correlation coefficient ($r_{max}$), describing how observational data at point scale correlates with observational data at pixel scale, which is particularly low for the two-parameter allometric equations (Fig. 5). Albeit the fact that three out of four DGVMs achieve a relatively good agreement between simulated and observed patterns at the pixel scale we find substantial uncertainty in the observational data due to spatial heterogeneity of local vegetation characteristics such as the structural and functional tree species composition

affecting biomass estimates across the Amazon (see also Rödig et al. 2017). The uncertainty resulting from conversion of raw inventory measurements into biomass from different allometric equations is generally neglected in model-data comparisons. However, it strongly affects our pixel-to-point comparison metrics, thereby remaining an important bottleneck for good model-data biomass comparisons. We suggest to include AGB estimates with associated uncertainties, e.g. using Bayesian inference procedures (see Réjou-Méchain et al., 2017) or to directly compare modelled allometries and related parameters in DGVMs with observational data.

## 4.3 What can we learn from including spatial heterogeneity and underlying drivers of biomass?

While our approach shows that some models could provide robust estimates for standing biomass stocks across the Amazon region (c.f. Fig. 2) it highlights that currently DGVMs do not represent productivity and related turnover correctly (i.e. the relation between productivity and residence time of woody biomass). As a result, the models might simulate the correct patterns for the wrong reasons as far as it can be derived from observational data. The four DGVMs applied in this study generally capture the observed pattern of AGB but strongly overestimate observed WP and underestimate $\tau$, and, from a pixel perspective, do not show strong variability across the Amazon region, thereby not capturing observed gradients (c.f. Fig. 6 and Tab. 1). WP and $\tau$ are driving AGB and are calculated by different schemes in the four DGVMs, e.g., regarding carbon allocation and drivers of mortality. Ground observations suggest that forest structure, forest dynamics and species composition vary across the Amazon region, such that variations in geology and soil fertility/mechanical properties coincide with region-wide variations in aboveground biomass, growth and stem mortality rates (Johnson et al., 2016;Quesada et al., 2012). In accordance, recent studies highlight that variation in stem mortality rates determines spatial variation in AGB and conclude that mortality should be modeled on the basis of individual stems, since stem-size distributions and stand density are important for predicting variation in aboveground biomass (Johnson et al., 2016;Rödig et al., 2017;Pillet et al., 2018). Projected increasing disturbances with different sizes and frequency may be an important additional driver for further variations under future scenarios (Espírito-Santo et al., 2014;Rödig et al., 2017). Nonetheless, the mechanisms leading to stem mortality need to be implemented in models based on experimental data that are only recently becoming available (Meir et al., 2015;Rowland et al., 2015). Overall, the DGVMs are able to reproduce the observed spatial pattern of AGB across the Amazon region, whereas for WP the model performance is less good and reproduction of the spatial pattern in mortality is generally very poor (Fig. 2, 6, 7). This suggests that models need to account for processes such as WP and mortality more mechanistically by including factors associated with resource limitation and disturbances regimes (see also Johnson et al. 2016). Recent efforts aiming at improving simulated Amazon forest biomass and productivity by including spatial variation in biophysical parameters (such as $\tau$ and $Vc_{max}$) have found that using single values for key parameters limits simulation accuracy (Castanho et al., 2013). Thus, we conclude that a more mechanistic representation of the processes driving the spatial variability of carbon stocks and fluxes, forest structure and tree demographic dynamics is necessary to improve simulation accuracy (Rödig et al., 2018).

## 5 Future applications of the methodological approach and outlook

In general, we assume that the basic concept of our method is applicable to any comparison between two datasets that are characterized by differences in spatial scale. If the process that causes small-scale variability can be approximated as white noise, corrected statistics can be computed. Notwithstanding future developments of next generation DGVMs, the most

relevant step of the presented approach is to account for the within-pixel variability $\sigma_\varepsilon$ from the point data to allow for a comparison of observational and simulated data. Due to relatively sparse plot data availability, we assume here that $\sigma_\varepsilon$ is stationary across the Amazon region. To evaluate this assumption further, we have calculated a regional within-pixel variability $\sigma_\varepsilon$ (Fig. S2) and find that it is in the range of the Amazon-wide $\sigma_\varepsilon$ of 28 to 36 Mg C ha$^{-1}$ (depending on the allometric equation used, see Tab. S2). Field studies show that forest dynamics vary locally, mostly due to variations in natural disturbance regimes,

mortality and edaphic properties (e.g. Baker et al., 2004;Chambers et al., 2013;Chave et al., 2006;Malhi et al., 2006;John et al., 2007), which in turn strongly influences our calculated within-pixel variability and thus, the metrics of the pixel-to-point comparison. Recent regional studies, that combine observational plot data and remote-sensing products from applications such as LIDAR (regions of Peru: Marvin et al. (2014), French Guyana: Fayad et al. (2016), Congo: Xu et al. (2017)), have already proven to detect spatial variability at high spatial resolution, which could be used to calculate a pixel-wise within-pixel

variability based on our approach. Upcoming remote sensing missions as the Global Ecosystem Dynamics Investigation Lidar (GEDI), the ESA BIOMASS mission, the NASA-ISRO Synthetic Aperture Radar (NISAR) mission, or the proposed Tandem-L mission (Moreira et al., 2015) will have the potential to provide non-stationary values of within-pixel variability for all regions of the Amazon. Thus, it is desirable to include regionally or locally specific estimates of $\sigma_\varepsilon^2$ in our analyses, which could be derived e.g. from above mentioned remote sensing data or from individual tree-based high-resolution simulations

(e.g. Rödig et al., 2017). In any case, we conclude that upcoming model-data comparison studies should at least account for stationary within-pixel variability when comparing simulated spatial data to data from discrete observational networks.

## 6 Code and data availability

All models are described in more detail in the supplementary material. The model code for LPJmL is available at https://github.com/PIK-LPJmL/LPJmL and archived under https://doi.org/10.5880/pik.2018.002. The model code for

ORCHIDEE is available at http://dx.doi.org/10.14768/06337394-73A9-407C-9997-0E380DAC5597. The model code for JULES is available from the JULES FCM repository: https://code.metoffice.gov.uk/trac/jules (registration required). The model code for INLAND is available at http://www.ccst.inpe.br/wp-content/uploads/inland/inland2.0.tar.gz. The permanent archive of the observational data from Mitchard et al. (2014) can be accessed at http://dx.doi.org/10.5521/FORESTPLOTS.NET/2014_1, see also Lopez-Gonzales et al. (2014). The inventory data from

Brienen et al. (2015) are available at http://dx.doi.org/10.5521/ForestPlots.net/2014_4, see also Brienen et al. (2014).

## 7 Author contribution

AR and JH conceived the ideas and designed methodology, and contributed equally to the paper; AR, JH, ER, PP and CZ analysed the data; FL, KT, MG, CR, BC, GS performed simulation runs; all authors contributed to writing the paper.

## 8 Acknowledgements

We acknowledge funding from the European Union's Seventh Framework Programme AMAZALERT project (282664), the Helmholtz Alliance "Remote Sensing and Earth System Dynamics", and the Belmont Forum/BMBF funded project CLIMAX. We acknowledge the efforts of the TEAM, RAINFOR and ATDM projects making the observational datasets available.

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

**Figures**

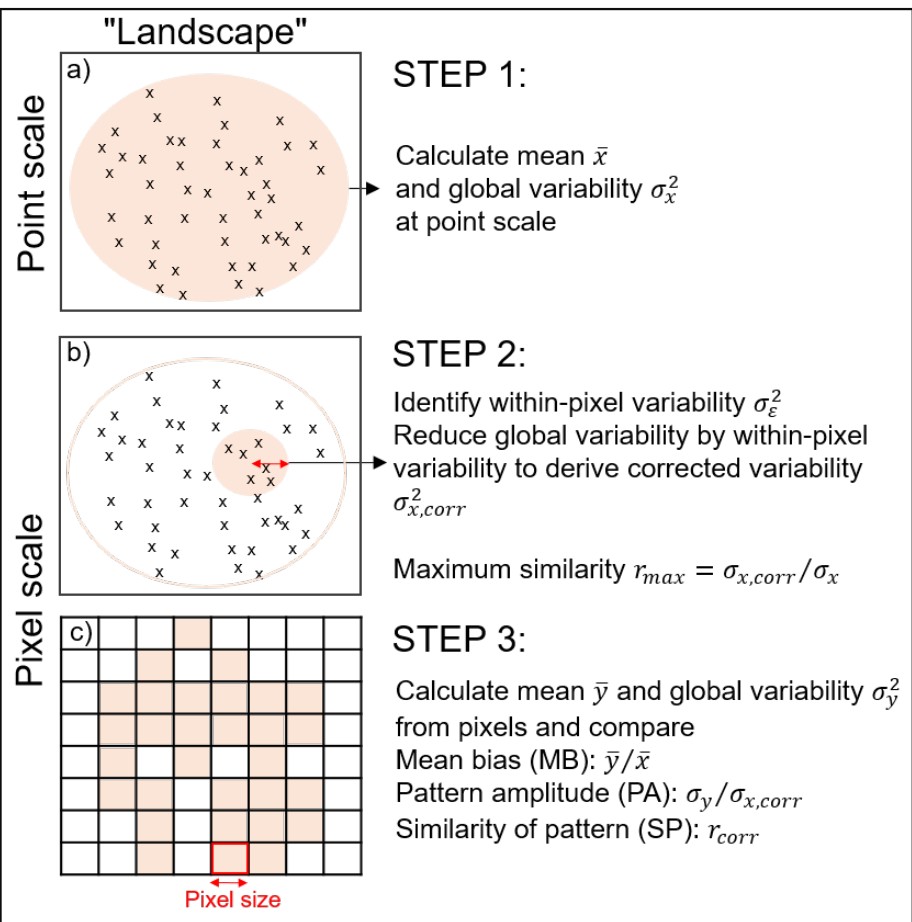

Figure 1: Schematic overview of the three steps in the pixel-to-point comparison. Note that we refer to "landscape" as the region of interest for which point and pixel data are available. (a) the mean and global variability at point scale are calculated from all plots across the landscape; (b) the within-pixel variability is calculated from all plots within the distance of the pixel-size (i.e. the red arrow corresponds with pixel size); (c) the mean and global variability is calculated from the pixel values and the three comparison metrics are derived. Note: A block bootstrapping with 10000 repetitions is performed to derive confidence intervals of the comparison metrics. The detailed set of equations to calculate maximum similarity, PA and SP can be found in the SI.

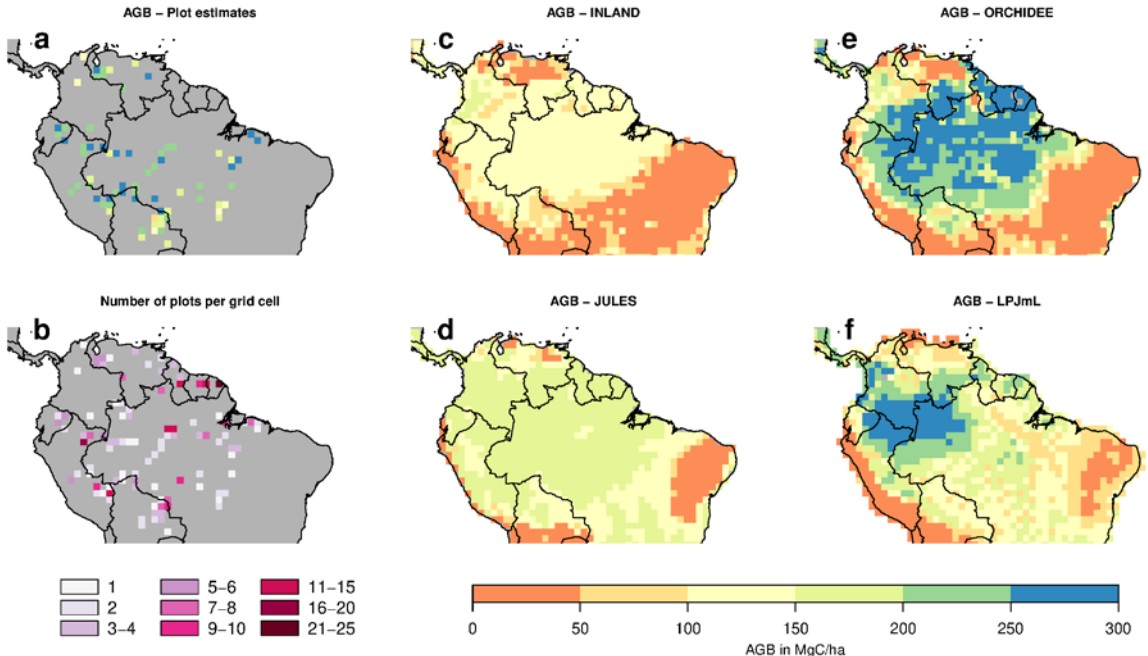

**Figure 2. Estimates of aboveground biomass (AGB) from forest plots in 1° x 1° pixels. (a) Mean AGB per pixel derived from inventory data based on one allometric equation ($K_{DH\rho}$, see SI for explanation and other allometric equations). (b) Number of plots per pixel and (c-f) simulated AGB from four DGVMs.**

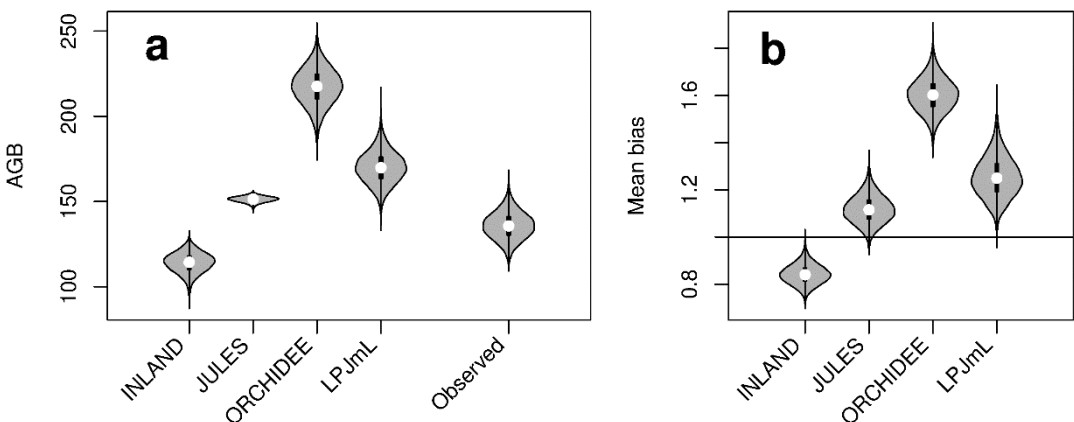

**Figure 3. Distribution of aboveground biomass (AGB in MgC/ha) from the four DGVMs and from the observational plots (see also Table S2 and S3). The figure shows (a) the mean value (white dot) and distribution from bootstrapping of absolute AGB values from the four simulations and observed (grey violins). b) Mean bias as the ratio of mean simulated and mean observed AGB ($\bar{y}/\bar{x}$).**

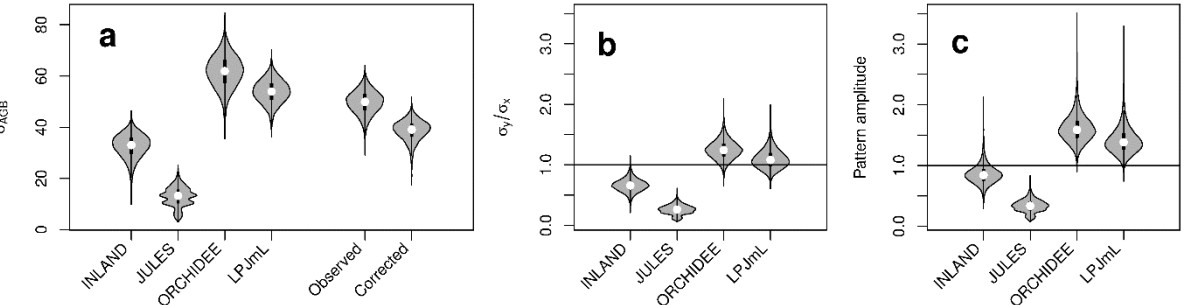

**Figure 4: (a) Standard deviations of AGB (in MgC/ha) for the four models and observational data (for the other allometric equations see also Table S2, S3). For the observational data, the global variability at point scale ("observed") and the corrected variability at pixel scale ("corrected") is given; (b) ratio of standard deviations without correcting for within-pixel variability, (c) corrected metrics of pattern amplitude ($\sigma_y/\sigma_{x,corr}$).**

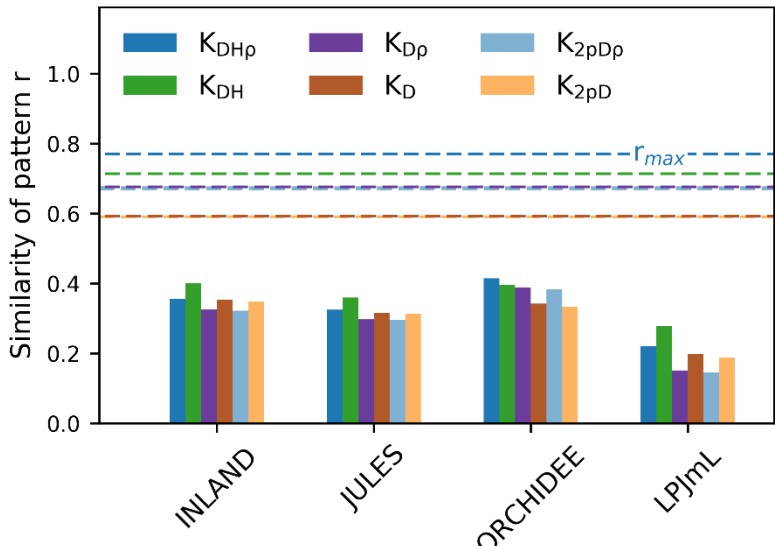

**Figure 5: The similarity of the observed vs. simulated spatial pattern of AGB at pixel scale (as indicated by *r* given in bars). The similarity is calculated for different versions of observed AGB derived from six allometric equations (indicated by the colors, see Tab. S1). The dashed line shows the maximum achievable correlation coefficients $r_{max}$ from the observational data and for the different allometric equations.**

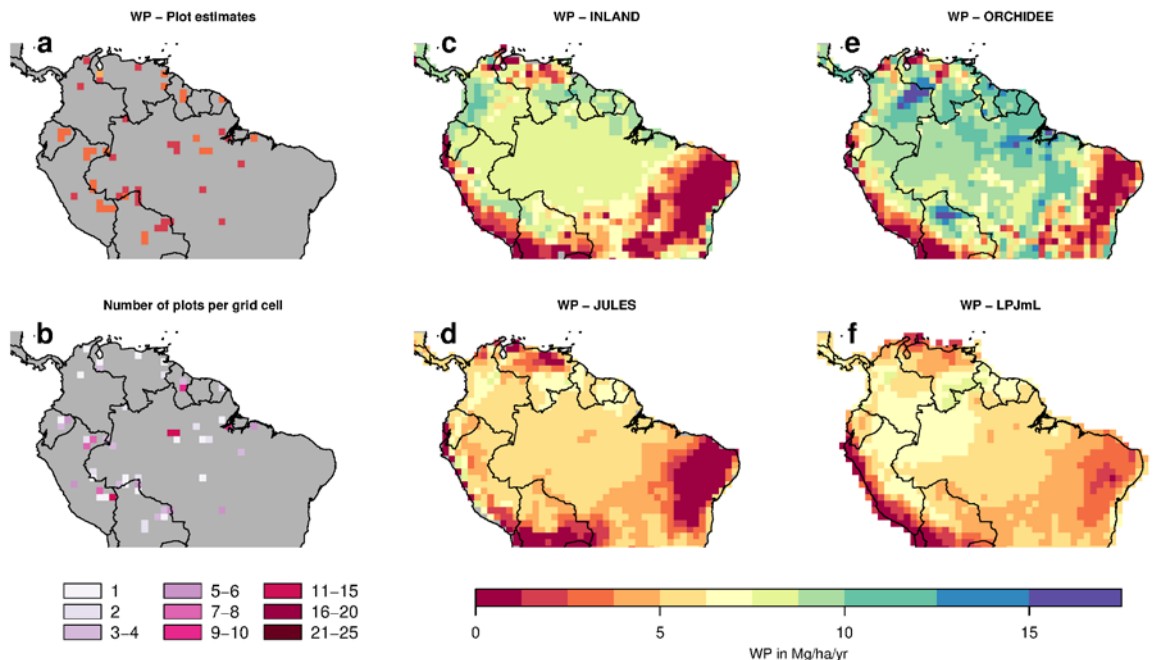

**Figure 6. Estimates of aboveground woody productivity (WP) from forest plots in 1° x 1° pixels. (a) Mean WP from inventory plots. (b) Number of plots per pixel and (c-f) simulated WP from four DGVMs.**

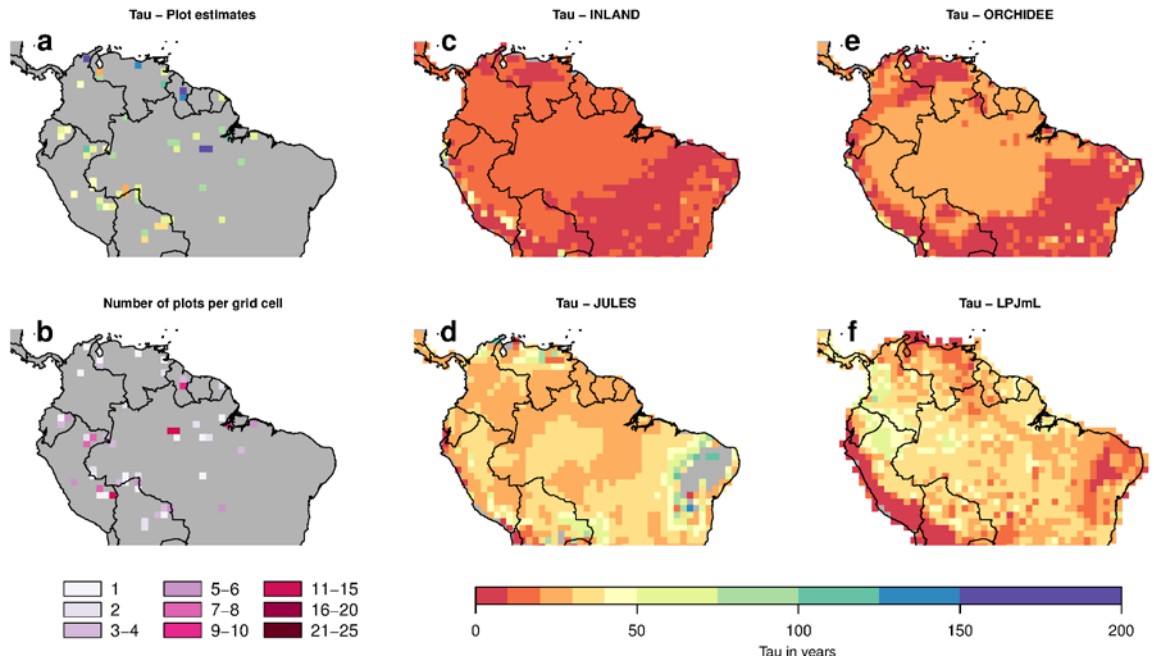

**Figure 7. Estimates of woody biomass residence time (τ) from forest plots in 1° x 1° pixels. (b) Mean residence time from inventory plots. (b) Number of plots per pixel and (c-f) simulated τ from four DGVMs.**

**Tables**

**Table 1. Results of the point-to-pixel comparison for (A) woody productivity (WP) and (B) residence time of woody biomass (τ). In brackets, the 5% and 95% confidence intervals are given. Blue boxes indicate when models overestimate observed values, red boxes indicate underestimation.**

| A) Woody productivity (WP) | Mean $\overline{x}$ (Mg/ha/yr) | Corrected global variability $\sigma_{x,corr}$ (Mg/ha/yr) | Max. achievable correlation $r_{max}$ |
|---|---|---|---|
| Observed | 2.57 | 0.38 | 0.67 |
| | Mean bias $(\overline{y}/\overline{x})$ | Pattern amplitude $(\sigma_y/\sigma_{x,corr})$ | Similarity of pattern $(r_{corr})$ |
| INLAND | 3.11 (2.91 – 3.31) | 2.91 (1.75 – 4.83) | 0.36 (0.11 – 0.35) |
| JULES | 2.01 (1.88 – 2.14) | 1.91 (1.08 – 3.25) | 0.38 (0.07 – 0.37) |
| ORCHIDEE | 3.55 (3.21 – 3.96) | 4.26 (2.64 – 6.96) | 0.03 (-0.25 – 0.01) |
| LPJmL | 1.74 (1.63 – 1.83) | 1.99 (1.36 – 3.16) | 0.50 (0.27 – 0.50) |
| B) Residence time (τ) | $\overline{x}$ (years) | $\sigma_{x,corr}$ (years) | $r_{max}$ |
| Observed | 73.84 | 28.04 | 0.64 |
| | Mean bias $(\overline{y}/\overline{x})$ | Pattern amplitude $(\sigma_y/\sigma_{x,corr})$ | Similarity of pattern $(r_{corr})$ |
| INLAND | 0.20 (0.17 – 0.24) | 0.13 (0.06 – 0.25) | 0.01 (-0.30 – 0.02) |
| JULES | 0.42 (0.35 – 0.51) | 0.24 (0.07 – 0.46) | -0.23 (-0.68 – -0.22) |
| ORCHIDEE | 0.35 (0.29 – 0.42) | 0.24 (0.12 – 0.45) | 0.08 (-0.22 – 0.08) |
| LPJmL | 0.47 (0.38 – 0.59) | 0.35 (0.19 – 0.61) | -0.18 (-0.46 – -0.18) |