# Peer review of "A generic pixel-to-point comparison for simulated large-scale ecosystem properties and ground-based observations: an example from the Amazon region"

_Geoscientific Model Development, 2018_

## Short Comment (SC1) · 31 Jul 2018

Dear authors,

In my role as Executive editor of GMD, I would like remind you of our Editorial, which states that "All papers must include a section, at the end of the paper, entitled 'Code availability'. Here, either instructions for obtaining the code, or the reasons why the code is not available should be clearly stated. It is preferred for the code to be uploaded as a supplement or to be made available at a data repository with

an associated DOI (digital object identifier) for the exact model version described in the paper. Alternatively, for established models, there may be an existing means of accessing the code through a particular system. In this case, there must exist a means of permanently accessing the precise model version described in the paper. In some cases, authors may prefer to put models on their own website, or to act as a point of contact for obtaining the code. Given the impermanence of websites and email addresses, this is not encouraged, and authors should consider improving the availability with a more permanent arrangement. After the paper is accepted the model archive should be updated to include a link to the GMD paper."

Thus please add for all used models explicitly how to access the code.

Additionally, the reference of Brienen et al. 2014 is unfortunately not part of the reference list.

Yours,

Astrid Kerkweg
* * *
─────────────────────────

---

## Referee Comment (RC1) · Anonymous Referee #1 · 18 Sep 2018

I have found this paper quite interesting as how to approach scale mismatch. The mathematical/statistical procedures are rigorous, and appropriate to derive indexes.

I only suggest including brief comments or modifying the following:

Do you think that DGVM models might increase resolution by using remote sensing data in order to improve spatial variability? Such as MODIS GPP/NPP at 1km.

I see more emphasis in spatial variability than temporal variability. How to assess the variability of the observed/estimated forest properties (as biomass) due to gap

dynamics? Is this more important than edaphic properties or Site Index?

Do you consider that some of the forest plots you used were subject to selective logging or natural gaps? and for this reason a cause of mismatch.

Can you provide a reference of studies calculating a "pixel-wise within-pixel variability"? See line 1 page 10.

Page 2, Line 27, do you mean "uneven spatial distribution and -temporal- variability due to natural gap dynamics"

Page 3, line 10, there is a typo error.

Page 3 line 24, what do you mean by "resolution of the study area"

I found figure 1 misleading. At some point, the distance between plots (in b) should depict what "corresponds to the size of the grid cell" (in c). Or maybe explaining this better in the caption.

Page 5, line 30. I assume is 2.1.3 instead of 1.3
* * *

---

## Referee Comment (RC2) · Anonymous Referee #2 · 28 Sep 2018

The present paper addresses an important problem in model data comparisons, namely how to compare measurements at the plot scale with pixel level model predictions, given the well known spatial heterogeneity in the measured variables. The authors propose a straightforward statistical framework that can take into account within pixel variations. The authors exemplify the use of this model using above-ground biomass(AGB) measurements and model predictions in the Amazon region. They show that by using the new metric for data variation it is easier to show whether the model predictions match the observed variability.

This core part of the paper is very clear and, in my opinion, very valuable for future modelling studies. The paper in general is very well written.

There are however some parts of the paper that are not so well developed, in particular the model comparison with the other datasets. In my opinion, the authors can go one of two ways: either cut down to the core of their method and the AGB data, or extend the less well developed and explained parts of their paper.

**Detailed comments**

**Description of observed data (section 2.2.1)** I find the data description in the main text of the paper extremely short. While the concept of AGB might be widely used and easily understandable, woody productivity and woody loss are not and a brief definition and description of how these were measured/calculated would greatly help the reader understand the subsequent analyses.

**Analysis of the woody productivity and loss.** The AGB data and mode predictions are analysed in detail and presented in four different figures, while the other two datasets have one figure each and one joint table. I would find it interesting to see a bit more detail about these observations too, especially since one of the strong discussion points (section 4.3) revolves around the model's inability to predict productivity and loss.

**Different allometric models.** While i fundamentally understand why the choice of allometric model is important for estimates of AGB, it does not feel like this additional dimension adds to the central message of the study. Mot of the detail fo the allometric models is buried in the supplementary material an, as far as I understand, the majority of the analysis has been done with only one of the allometric equations.

---

## Author Response (AR1)

Dear Editors,

Please find below a point-by-point response to all referee comments specifying all changes in the revised manuscript. The revised manuscript including changes marked in red can be found at the end of this document.

On behalf of the co-authors,

Anja Rammig

**Response to Executive Editor Comment:**

In my role as Executive editor of GMD, I would like remind you of our Editorial, which states that "All papers must include a section, at the end of the paper, entitled 'Code availability'. Here, either instructions for obtaining the code, or the reasons why the code is not available should be clearly stated. It is preferred for the code to be uploaded as a supplement or to be made available at a data repository with an associated DOI (digital object identifier) for the exact model version described in the paper. Alternatively, for established models, there may be an existing means of accessing the code through a particular system. In this case, there must exist a means of permanently accessing the precise model version described in the paper. In some cases, authors may prefer to put models on their own website, or to act as a point of contact for obtaining the code. Given the impermanence of websites and email addresses, this is not encouraged, and authors should consider improving the availability with a more permanent arrangement. After the paper is accepted the model archive should be updated to include a link to the GMD paper." Thus please add for all used models explicitly how to access the code. Additionally, the reference of Brienen et al. 2014 is unfortunately not part of the reference list.

*Response: Thank you for your comment. We have included in section 6: "All models are described in more detail in the supplementary material. The model code for LPJmL is available through PIK's gitlab server at https://gitlab.pik-potsdam.de/lpjml/LPJmL and archived under https://doi.org/10.5880/pik.2018.002. The model code for ORCHIDEE is available under http://dx.doi.org/10.14768/06337394-73A9-407C-9997-0E380DAC5597. The model code for JULES is*

*available from the JULES FCM repository: https://code.metoffice.gov.uk/trac/jules (registration required). The model code for INLAND is available from http://www.ccst.inpe.br/wp-content/uploads/inland/inland2.0.tar.gz. The permanent archive of the observational data from Mitchard et al. (2014) can be accessed at http://dx.doi.org/10.5521/FORESTPLOTS.NET/2014_1, see also Lopez-Gonzales et al. (2014). The inventory data from Brienen et al. (2015) are available at http://dx.doi.org/10.5521/ForestPlots.net/2014_4, see also Brienen et al. (2014)."*

*We have included the reference to Brienen et al. 2014 in the reference list.*

**Response to Referee Comment #1:**

I have found this paper quite interesting as how to approach scale mismatch. The mathematical/statistical procedures are rigorous, and appropriate to derive indexes.

*Response: Thank you for the very positive evaluation of our manuscript. Below we provide a point-to-point reply to your comments.*

I only suggest including brief comments or modifying the following:

Do you think that DGVM models might increase resolution by using remote sensing data in order to improve spatial variability? Such as MODIS GPP/NPP at 1km.

*Response: The resolution of DGVMs is always defined by the resolution of the climate input data. Spatial variability in DGVMS could hence only be increased by using input climate data at higher resolution (e.g., from regional climate models). We have explained this in more detail on p.6, l.5. As simulated GPP and NPP are a model output, MODIS GPP and NPP could only be used to evaluate model output. However, note that MODIS is derived from NDVI which tends to saturate in dense forests such as the Amazon and, thus, has limited capability to identify spatial variations (Hall et al 2011). As pointed out in the conclusion (page 9, line 30) the integration of forest structural information derived from upcoming Lidar remote sensing missions (such as GEDI or BIOMASS) may reveal more information on spatial variability than previous optical remote sensing (NDVI).*

I see more emphasis in spatial variability than temporal variability. How to assess the variability of the observed/estimated forest properties (as biomass) due to gap dynamics? Is this more important than edaphic properties or Site Index?

*Response: Thank you for pointing this out. Gap dynamics clearly have an effect on the spatial variability of biomass (e.g. see Rödig et al. 2017, 2018) and this is discussed in our manuscript on page 9, l.15-18. However, we emphasise that most gaps in tropical forests are much smaller than the one hectare scale of the plot data used for model validation (e.g. 99% of gaps on BCI are <0.04 ha; Hubbell et al. 1999), and so the plots are generally representative of the disturbance regime in each landscape. Additionally, gap dynamics and site properties cannot be separated easily as site characteristics strongly influence forest dynamics (e.g. Quesada et al. 2011). We have made this more clear in the introduction by rephrasing p.2, l.18-22. We agree that it would be interesting to assess the temporal variability and the respective model behaviour, but this is out of the scope of this study.*

Do you consider that some of the forest plots you used were subject to selective logging or natural gaps? and for this reason a cause of mismatch.

*Response: Thank you for pointing this out. The plot data are taken from Brienen et al. (2014, 2015) and Mitchard et al. (2014) who excluded all plots that were subject to anthropogenic disturbances, including selective logging, so this is not a cause of the mismatch. As above, the spatial scale of the plot data means that they incorporate most size classes of natural gaps, particularly as the plot data were averaged across sites occurring within the same pixel. Across the plot network, the biases introduced into estimate of carbon balance by one hectare plots not sampling the very largest and rarest natural gaps are in fact very small (Espirito-Santo et al. 2014). We have revised on p.5 the section "2.2.1 Observed data at point scale: Description of site-level data".*

Can you provide a reference of studies calculating a "pixel-wise within-pixel variability"? See line 1 page 10.

*Response: To our knowledge, there is no previous study that calculates pixel-wise within-pixel variability. This is the reason why we propose this in our manuscript. We have revised the text in l. 1, p.10 to "… which could be used to calculate a pixel-wise within-pixel variability based on our approach." to make this more clear.*

Page 2, Line 27, do you mean "uneven spatial distribution and -temporal- variability due to natural gap dynamics"

*Response: We mean "uneven spatial distribution and spatial variability due to natural gap dynamics" and have changed it accordingly in the manuscript.*

Page 3, line 10, there is a typo error.

*Response: Thank you, is corrected.*

Page 3 line 24, what do you mean by "resolution of the study area"

*Response: Thank you, this is indeed not fully clear in our text. We have revised the sentence to "Landscape variability depends on the extent and heterogeneity of the study area (Turner et al., 2001). Point measurements within a pixel of larger spatial scale, for example, may reveal small-scale spatial variabilities within the pixel."*

I found figure 1 misleading. At some point, the distance between plots (in b) should depict what "corresponds to the size of the grid cell" (in c). Or maybe explaining this better in the caption.

*Response: Yes, we agree that Figure 1b is misleading. We have revised Figure 1 so that the red arrows indicate better that they correspond with the size of the grid cell and we have expanded the caption of Figure 1.*

Page 5, line 30. I assume is 2.1.3 instead of 1.3

*Response: Yes, agreed, we have corrected that.*

**Response to Referee Comment #2:**

The present paper addresses an important problem in model data comparisons, namely how to compare measurements at the plot scale with pixel level model predictions, given the well-known spatial heterogeneity in the measured variables. The authors propose a straightforward statistical framework that can take into account within pixel variations. The authors exemplify the use of this model using aboveground biomass (AGB) measurements and model predictions in the Amazon region. They show that by using the new metric for data variation it is easier to show whether the model predictions match the observed variability.

This core part of the paper is very clear and, in my opinion, very valuable for future modelling studies. The paper in general is very well written. There are however some parts of the paper that are not so well developed, in particular the model comparison with the other datasets. In my opinion, the authors

can go one of two ways: either cut down to the core of their method and the AGB data, or extend the less well developed and explained parts of their paper.

*Response: Thank you very much for the positive feedback on our manuscript. We appreciate the suggestion for improving the part of the manuscript about the model comparison with other datasets and have expanded it. Please see below our response to your detailed comments.*

Detailed comments

Description of observed data (section 2.2.1) I find the data description in the main text of the paper extremely short. While the concept of AGB might be widely used and easily understandable, woody productivity and woody loss are not and a brief definition and description of how these were measured/calculated would greatly help the reader understand the subsequent analyses.

*Response: Yes, we agree and we thank the reviewer for pointing this out. We have described the observational data in more detail in section 2.2.1, in particular how woody productivity and woody loss were measured and calculated. Brienen et al. (2015) derived "…forest woody productivity… from the sum of biomass growth of surviving trees and trees that recruited (that is, reached a diameter ≥ 100 mm),and mortality [=woody loss] from the biomass of trees that died between censuses. We have added this accordingly in our methods section on p. 5, sect. 2.2.1*

Analysis of the woody productivity and loss. The AGB data and mode predictions are analysed in detail and presented in four different figures, while the other two datasets have one figure each and one joint table. I would find it interesting to see a bit more detail about these observations too, especially since one of the strong discussion points (section 4.3) revolves around the model's inability to predict productivity and loss.

*Response: Thank you for your comment. We agree that woody productivity and loss are also interesting. However, we decided to focus on mainly AGB in the main manuscript as an example to present our approach. We prefer to keep the detailed figures in the supplementary material.*

Different allometric models. While I fundamentally understand why the choice of allometric model is important for estimates of AGB, it does not feel like this additional dimension adds to the central message of the study. Most of the detail for the allometric models is buried in the supplementary

material and, as far as I understand, the majority of the analysis has been done with only one of the allometric equations.

*Response: Thank you for pointing this out. However, we feel that the uncertainty introduced by allometric relations for the comparison of field observations and DGVM simulations is an aspect that needs to be part of the discussion of our paper. We have now added more detail on the allometric equations in the methods section (p. 5, sect. 2.2.1) and we have improved the discussion about the relevance of the allometric equations on p. 8 and 9.*

[revised manuscript text omitted]